

# Antimicrobial resistant gene prevalence in soils due to animal manure deposition and long-term pasture management

Yichao Yang[1,*], Amanda J. Ashworth[2,*], Jennifer M. DeBruyn[3], Lisa M. Durso[4], Mary Savin[1], Kim Cook[5], Philip A. Moore Jr.[2] and Phillip R. Owens[6]

[1] Department of Crop, Soil, and Environmental Sciences, University of Arkansas at Fayetteville, Fayetteville, AR, United States of America

[2] Poultry Production and Product Safety Research Unit, United States Department of Agriculture, Agricultural Research Service, Fayetteville, AR, United States of America

[3] Department of Biosystems Engineering & Soil Science, University of Tennessee - Knoxville, Knoxville, TN, United States of America

[4] Agroecosystem Management Research Unit, United States Department of Agriculture, Agricultural Research Service, Lincoln, NE, United States of America

[5] United States Department of Agriculture, Agriculture Research Service, Beltsville, MD, United States of America

[6] Dale Bumpers Small Farms Research Center, United States Department of Agriculture, Agricultural Research Service, Booneville, AR, United States of America

[*] These authors contributed equally to this work.

Corresponding author
Amanda J. Ashworth,
Amanda.Ashworth@usda.gov

## ABSTRACT

The persistence of antimicrobial resistant (AMR) genes in the soil-environment is a concern, yet practices that mitigate AMR are poorly understood, especially in grasslands. Animal manures are widely deposited on grasslands, which are the largest agricultural land-use in the United States. These nutrient-rich manures may contain AMR genes. The aim of this study was to enumerate AMR genes in grassland soils following 14-years of poultry litter and cattle manure deposition and evaluate if best management practices (rotationally grazed with a riparian (RBR) area and a fenced riparian buffer strip (RBS), which excluded cattle grazing and poultry litter applications) relative to standard pasture management (continuously grazed (CG) and hayed (H)) minimize the presence and amount of AMR genes. Quantitative PCR (Q-PCR) was performed to enumerate four AMR genes (*ermB*, *sulI*, *intlI*, and *bla*$_{ctx-m-32}$) in soil, cattle manure, and poultry litter environments. Six soil samples were additionally subjected to metagenomic sequencing and resistance genes were identified from assembled sequences. Following 14-years of continuous management, *ermB*, *sulI,* and *intlI* genes in soil were greatest ($P < 0.05$) in samples collected under long-term continuous grazing (relative to conservation best management practices), under suggesting overgrazing and continuous cattle manure deposition may increase AMR gene presence. In general, AMR gene prevalence increased downslope, suggesting potential lateral movement and accumulation based on landscape position. Poultry litter had lower abundance of AMR genes (*ermB*, *sulI*, and *intlI*) relative to cattle manure. Long-term applications of poultry litter increased the abundance of *sulI* and *intlI* genes in soil ($P < 0.05$). Similarly, metagenomic shotgun sequencing revealed a greater total number of AMR genes under long-term CG, while fewer AMR genes were found in H (no cattle manure) and RBS (no animal manure or poultry litter). Results indicate long-term

conservation pasture management practices (e.g., RBS and RBR) and select animal manure (poultry litter inputs) may minimize the presence and abundance of AMR genes in grassland soils.

## INTRODUCTION

Veterinary pharmaceutical usage is a fundamental component of conventional poultry and bovine production for treating microbial infections and increasing weight gains (*Collignon et al., 2009*). Repeated use of antibiotics during food-animal production may provide selection pressure for the evolutionary phenomenon known as antimicrobial resistance (AMR). Genes encoding resistance to antimicrobials and antibiotics, which can also naturally be found in many bacteria, can be transferred between organisms via horizontal gene transfer (*Juhas, 2015*). Agricultural practices influence the prevalence and occurrence of AMR genes in soils. For example, soils amended with cattle manure not treated with antibiotics contained higher abundance of $\beta$-lactam resistant bacteria than soils with inorganic fertilizer inputs (*Udikovic-Kolic et al., 2014*). In another study, soil applications of swine manure increased erythromycin resistance gene abundance and remained high for a decade post-application (*Scott et al., 2018*). However, AMR genes occur naturally(*Durso, Miller & Wienhold, 2012*); for example, *Cadena et al. (2018)* identified tetracycline and sulfonamide resistant genes in organic farms without routine antibiotic usage.

There is recent interest in monitoring the dissemination of AMR genes into the environment, particularly those directly relevant to human and animal health, as consumers and producers are increasingly concerned about antibiotic resistance in food systems (*Durso, Miller & Wienhold, 2012*). One goal of sustainable agriculture is to close nutrient cycles by applying animal manures to neighboring cropping systems. Additionally, depending on antibiotic properties, large quantities of undegraded antibiotics exit animals to manures, including poultry litter (a combination of bedding material and excreta); for example, up to 90% of sulfonamides and 25–75% of tetracyclines may be excreted into manure as the parent compound (*Kulshrestha, Giese & Aga, 2004*; *Thiele-Bruhn et al., 2004*). From manure, antibiotics, genes encoding AMR, and microbes may be transferred to soil (*Cook, Netthisinghe & Gilfillen, 2014*; *Heuer, Schmitt & Smalla, 2011*; *Zhang et al., 2017*; *He et al., 2014*). Therefore, cattle manure and poultry litter applications, which are valuable sources of nutrients such as N, P, and potassium (K), may also be a pathway for AMR bacteria and genes into the environment (*Yang et al., 2019a*).

The ability of pasture management practices (i.e., filter strips and rotational grazing) to reduce AMR gene presence, prevalence, and movement to soils is largely unknown. Our previous work indicated that continuously grazed systems increased soil microbial community richness and diversity owing to greater organic animal inputs (*Yang et al.,*

*2019b*), which suggests manure increases microbiome diversity and improves soil health. However, animal manure may also be a source for AMR genes. Therefore, the current work aims to understand the impacts of pasture management on AMR bacteria and gene presence. This study focused on quantifying four AMR associated genes (i.e., erythromycin resistance gene (*ermB*), sulfonamide resistance gene (*sulI*), integrase gene (*intII*), and *β*-lactam resistance gene (*bla_{ctx-m-32}*) present in pasture soil, cattle manure, and poultry broiler litter using Q-PCR in an effort to balance human, animal, and environmental priorities. These four genes are useful for understanding the ecology and biology of agricultural AMR genes in soil and manure systems (*Durso, Miller & Wienhold, 2012*). We additionally applied metagenomic sequencing to reveal the suite of resistance genes in the soil community and to evaluate best management practices that may reduce the presence of AMR genes from manure and poultry litter applications to the soil.

## MATERIALS AND METHODS

### Experimental design

In 2004, a field study was initiated by *Pilon et al. (2017a)*, *Pilon et al. (2017b)* and *Pilon et al. (2018)* at the USDA-ARS Unit in Booneville, Arkansas to evaluate how pasture management affects water quality. Nine watersheds (average slope of 8%) were constructed on Enders and Leadvale silt loams. Each watershed had a total area of 0.14 ha, with the dominant grass species being bermudagrass (*Cynodon dactylon* L.).

Briefly, three grazing strategies were implemented from 2004–2017 with three replications, including: continuously grazed (CG), hayed (H), and rotationally grazed with an ungrazed, fenced riparian strip (RBR; *Yang et al., 2019a*; *Yang et al., 2019b*) (Fig. 1). The CG treatment was consistently grazed by one to two calves during the year (*Pilon et al., 2017a*). The H treatment was hayed three times annually (April, June, and October) to a height of 10 cm (no cattle in these watersheds). The RBR system is considered a best management strategy and was rotationally grazed based on forage height (*Pilon et al., 2017a*; *Pilon et al., 2017b*; *Pilon et al., 2018*). Calves (three) were placed in rotationally grazed watersheds based on forage height (when heights were 20 to 25 cm) and removed (10 to 15 cm) (*Yang et al., 2019a*; *Yang et al., 2019b*). Each watershed was divided into 3 zones (perpendicular to slope) given that topography widely affects the microbial biogeography (*Yang et al., 2019a*; *Yang et al., 2019b*). Landscape positions corresponded to upper slope (zone 1), mid-slope (2), and downslope positions (3), whereas the RBR represented zone 4 (*Yang et al., 2019a*; *Yang et al., 2019b*). The riparian buffer strip (RBS) and served as the nested control. The length of the 3 zones in CG and H was 57 m and the length of the 3 zones in RBR was 42.75 m. Broiler litter was surface applied at 5.6 Mg dry matter per ha in April-May of each year per watershed (excluding the RBS). All poultry litter rates were equivalent on an aerial basis (*Yang et al., 2019a*; *Yang et al., 2019b*). Broiler litter was obtained annually from a nearby commercial broiler farm. The RBR watersheds had a fenced off, riparian area containing four tree species at the base of each watershed (*Pilon et al., 2017a*; *Yang et al., 2019a*; *Yang et al., 2019b*). The RBS (zone 4) was not grazed and did not receive poultry litter applications (Fig. 1). Watersheds received no other human or animal inputs during the project duration.
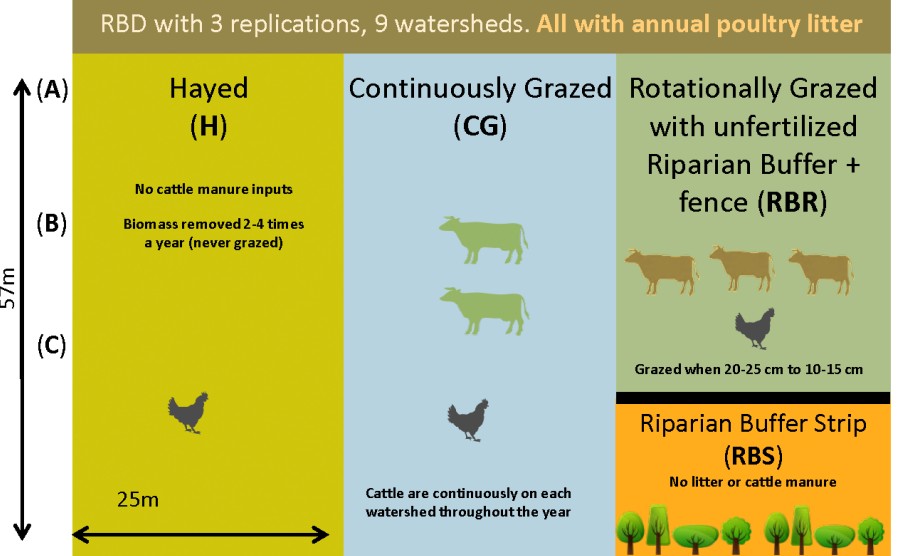

**Figure 1  Schematic representation of the experimental set-up.** Randomized complete block design with three replications (nine watersheds total) from 2004–2018. All areas have received annual poultry litter applications (except for the RBS area). The RBS received neither poultry litter nor cattle manure inputs. CG and RBR received cattle manure. Each watershed was divided, perpendicular to the slope into three zones (corresponding to shoulder (A), upper backslope (B), and lower backslope (C) positions), whereas the RBR consisted of these three zones plus the RBS (zone 4). (Soil samples, $n = 120$; cattle manure, $n = 12$; poultry litter, $n = 6$; and, shotgun sequencing, soil, $n = 6$).

## Sample collection

Soil sampling occurred once pre-poultry litter application (April) and other post-litter application (July; within 2 months following poultry litter land application) in 2016 and 2017 (four sampling dates total). Soils were manually collected to a 0–15 cm depth (6 randomly selected samples in zone centers and composited to one sample), with three replicates total ($n = 120$ total). One sample per zone was collected, transported in a cooler, and later stored at −80° C for DNA extraction (*Yang et al., 2019a*; *Yang et al., 2019b*). Sampling equipment was sterilized between watersheds with 70% ethanol ($C_2H_6O$).

Prior to poultry litter application on each watershed in 2018 and 2019, a central sample was collected from broiler house windrow piles ($n = 6$ total). In tandem, for treatments receiving cattle (Angus crosses) manure inputs (i.e., CG and RBR), two fresh manure samples were collected post deposition (within 24 h) per watershed ($n = 12$ total). Right before DNA extraction, fresh, composited samples (soil, manure, and poultry litter samples) were weighed and dried at 70 °C for 48 h and reweighed to determine gravimetric moisture content.

## Detection and analysis of four antibiotic resistance associated genes following long-term management using Q-PCR

Soil, cattle manure, and poultry litter samples were homogenized and 0.25 mg of each sample was used for DNA extraction. Isolation and purification of DNA from samples (soil: $n = 120$; cattle manure: $n = 12$; poultry litter: $n = 6$) was conducted with the PowerSoil DNA

**Table 1  Sequence and properties of the Q-PCR primers used in this project.**

| Target gene | Application/Primer | Sequence (5′–3′) | Annealing temperature (°C) | Amplicon size (bp) | Reference |
|---|---|---|---|---|---|
| *blactx-m-32* | ctx-m-32FWD<br>ctx-m-32RVS | CGTCACGCTGTTGTTAGGAA<br>CGCTCATCAGCACGATAAAG | 63 | 156 | *Szczepanowski et al. (2009)* |
| *ermB* | ermB-QPCR-F-Florez<br>ermB-QPCR-R-Florez | GGATTCTACAAGCGTACCTTGGA<br>AATCGAGACTTGAGTGTGCAAGAG | 60 | 69 | *Florez et al. (2014)* |
| *sulI* | sul I FW<br>sul I RVS | CGCACCGGAAACATCGCTGCAC<br>TGAAGTTCCGCCGCAAGGCTCG | 65 | 163 | *Barraud et al. (2010)* |
| *intlI* | intI1LC5 FW<br>intI1LC1 RV | GATCGGTCGAATGCGTGT<br>GCCTTGATGTTACCCGAGAG | 55 | 196 | *Pei et al. (2006)* |

Isolation Kit (MoBio Laboratories Inc., Cat. 12888-100) according to the manufacturer's protocol. Extracted DNA were quantified using Quant-iT™ PicoGreen™ dsDNA Assay Kit (ThermoFisher Scientific, Cat. P7589) and used directly in quantitative Q-PCR. All 120, 6, and 12 soil, cattle manure, and poultry litter DNA samples, respectively, were subjected to Q-PCR for detection of four genes associated with AMR as described in the clinical isolates, which included *ermB* (*Florez et al., 2014*), *sulI* (*Barraud et al., 2010*), *intlI* (*Pei et al., 2006*), and *bla$_{ctx-m-32}$* (*Szczepanowski et al., 2009*), using previously published primers (Table 1). Each PCR amplification was performed in triplicate. The positive control (named gBlock2 4G with 16S *ermB* Florez 1-18-17) is an 808bp double stranded synthetic gBlocks® gene fragment synthesized by Integrated DNA technologies, lnc. (*Blazejewski, Ho & Wang, 2019*). Standard curves consisted of a serial dilution (known gene fragment copy numbers ranging from $1.15 \times 10^5$ to $1.15 \times 10^{11}$ copies per 5 µl).

Amplifications were performed in a QuantStudio™ 3 Real-Time PCR system (ThermoFisher Scientific, Cat. A28137). Each 20 µL Q-PCR reaction included 5 µL of extracted DNA (approximately 100 ng) or standard, 10 µL of SYBR Green PCR Master Mix, and 100 mM of each primer. As a negative control, all sets of primers were tested with sterile water as the template; and all of them were below the threshold. Each reaction was technically replicated for three times per extracted sample DNA and standard DNA, resulting in an average cycle threshold (Ct) value used to estimate the initial quantity. Cycling conditions including annealing temperature specific for each gene are provided in Table 1. The amplification efficiency was between 92% and 105%, and the $R^2$ value was above 0.98. Baseline and threshold calculations were performed using QuantStudio®Design & Analysis software. Amplified products were visualized on a 1% agarose gel with an ethidium bromide stain. The quantities of gene copy numbers were then determined using standard curves. Gene copy abundances were then normalized per gram dry weight of soil, cattle manure, and poultry litter after measuring the moisture content of each sample. Finally, the gene copy numbers per gram dry weight were transformed into log10 values for further statistical analysis as they were not normally distributed (*Ganger, Dietz & Ewing, 2017*).

To detect significant differences for fixed effects (pasture management, sample collection timing, and zone) an analysis of variance (ANOVA) was conducted on log transformed data using JMP software (JMP®12; *SAS Institute Inc, 2014*) with replicate as a random effect. Probability values less than 0.05 were considered significant and pairwise posthoc comparisons were made using Tukey's Honestly Significant Difference test. Samples below detection limit were excluded in the analysis.

## Metagenomic sequencing and data analysis

To evaluate long-term effects of pasture management on AMR genes, metagenomic sequencing was applied for six soil samples (post-application zone 3, CG; post-application zone 3 H; post-application zone 3 RBR, post-application zone 3, RBS all replication 1; and, post-application zone 3 RBR, post-application zone 3, RBS replication 2). Sequencing libraries were prepared according to the Illumina Miseq sample preparation guide. Metagenomic sequencing was performed using Miseq Reagent Kits v2 with paired-end $2 \times 250$ bp reads on the MiSeq platform (University of Tennessee Genomics Core; Knoxville, TN). Raw data were uploaded to NCBI with the accession number of SAMN 14783573-14783584. Read trimming and assembly were performed according to *Tyson et al. (2015)*. Genomes were annotated using the AmrPlusPlus pipeline (*Lakin et al., 2017*). All samples resulted in a total of 5.19 Gb of sequence data. Trimmomatic was used for removal of low-quality bases and sequences (*Lakin et al., 2017*). Reads classified as host genome (*Bos taurus* and *Gallus gallus*) were removed from further analysis. The adapter contamination and low-quality reads were also removed. The database 'MEGARes' has been integrated inside the pipeline and used for identification of AMR genes. AMR genes with a gene fraction (i.e., proportion of nucleotides that aligned with at least one query read) of >85% coverage across all alignments were considered to be positively identified in a sample (*Noyes et al., 2016a*). The minimum length of a read was 150 and the mean Phred score was above 30. The AMR gene analysis was carried out using the Resistome Analyzer tool (https://github.com/cdeanj/resistomeanalyzer) (*Lakin et al., 2017*). Utilizing this tool, three annotation levels were produced, which include gene-level (sequencing-level), mechanism-level, and class-level counts.

# RESULTS

## Distribution of four AMR associated genes in soil based on pasture management, landscape position, and sampling time

Soil *ermB* gene abundance varied among treatments (CG, H, RBR, and RBS) and zones (1, 2, 3, and 4) ($P < 0.05$) (Fig. 2A), although, sample collection time (pre- or post-poultry litter applications; $P > 0.05$) had no impact on the abundance of *ermB* gene (Table 2; Fig. 2B). There was an interaction effect from pasture management and zone on the abundance of *ermB* genes. Across pasture management, the highest abundance of *ermB* gene was found in the CG treatment ($\mu$ log gene copies per gram dry weight soil $= 3.03$), followed by H (2.86 gene copies per gram dry weight soil), RBR (2.72 gene copies per gram dry weight soil) and RBS (0.73 gene copies per gram dry weight soil) (Fig. 2A). Compared with RBS, CG increased the abundance of *ermB* by 2.3 log, H increased the abundance of

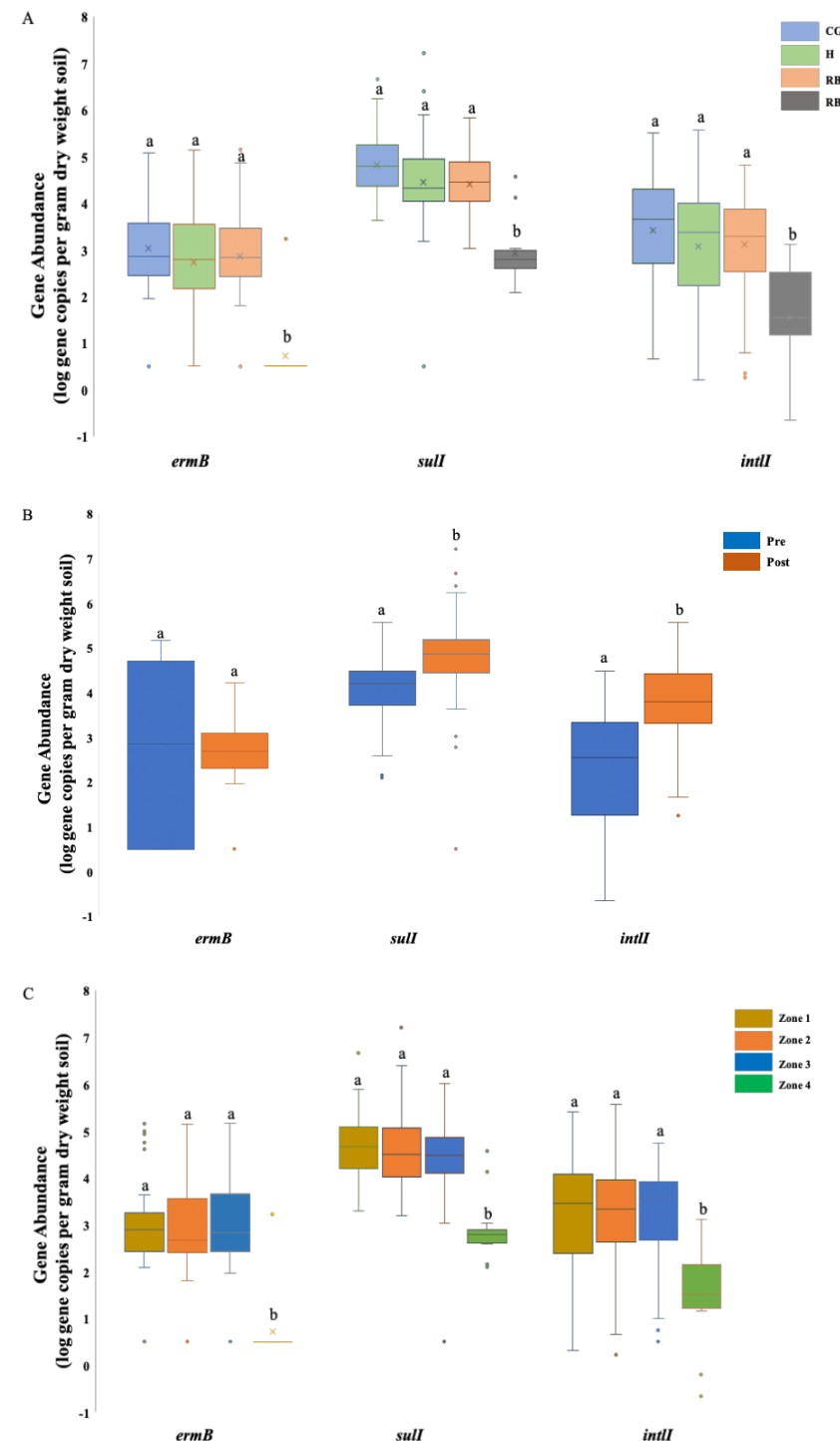

**Figure 2   ANOVA results illustrating the abundance of three AMR associated genes in cattle manure and poultry litter.** ANOVA results illustrating the differences of the abundance of three AMR associated genes were impacted by the single factor, animals (cattle manure vs. poultry litter), and year (2018 vs. 2019), and interaction between these two factors in cattle manure and poultry litter samples collected from 2018–2019. CG (continuously grazed), H (hayed), and RBR (rotationally grazed with a fenced off riparian buffer). Error bars represent standard errors. The connecting letters indicates a significant difference at an alpha level of 0.05. Mean abundances of three AMR associated genes, ermB, sulI and intlI amplified from soil genomic DNA samples based on (A) pasture management, (B) sampling time (pre and post poultry litter application) and (C) zone (zone 1, zone 2, zone 3, and zone 4).

**Table 2 Differences in abundance of three AMR genes.** ANOVA results testing for differences in abundance of three AMR genes (log gene copies per gram dry weight soil) in soil samples influenced by main factors, pasture management (CG, H, RBR and RBS), sampling time (pre and post) and zone (zone 1, zone 2, zone 3, and zone 4), as well as the interactions between two main factors (pasture $\times$ timing, pasture $\times$ zone, and timing $\times$ zone) and three main factors (pasture $\times$ timing $\times$ zone) at Booneville, AR from 2016-2017.

| Parameter | Factor | df | F-value | P-value |
|---|---|---|---|---|
| ermB | Pasture Management | 3 | 10.399 | <0.0001* |
| | Time | 1 | 2.464 | 0.119 |
| | Zone | 3 | 10.088 | <0.0001* |
| | Pasture × Time | 3 | 1.299 | 0.277 |
| | Pasture × Zone | 9 | 3.813 | <0.001* |
| | Zone × Time | 3 | 1.245 | 0.297 |
| | Pasture × Zone × Time | 9 | 0.366 | 0.949 |
| sulI | Pasture Management | 3 | 16.188 | <0.001* |
| | Time | 1 | 20.499 | <0.001* |
| | Zone | 3 | 14.917 | <0.001* |
| | Pasture × Time | 3 | 1.035 | 0.380 |
| | Pasture × Zone | 9 | 5.851 | <0.001* |
| | Zone × Time | 3 | 1.640 | 0.184 |
| | Pasture × Zone × Time | 9 | 1.532 | 0.145 |
| intlI | Pasture Management | 3 | 6.891 | <0.001* |
| | Time | 1 | 56.512 | <0.001* |
| | Zone | 3 | 6.268 | <0.001* |
| | Pasture × Time | 3 | 3.093 | <0.05* |
| | Pasture × Zone | 9 | 2.256 | <0.05* |
| | Zone x Time | 3 | 3.058 | <0.05* |
| | Pasture x Zone x Time | 9 | 1.553 | 0.139 |

Note
*$p$ <0.05

ermB by 2.13 log, and RB increased the abundance of ermB by 1.99 log. Among zones, the greatest abundance of ermB occurred in zone 3 (μ log gene copy numbers per dry weight = 2.92), followed by zone 2 (2.91 gene copy numbers per dry weight), zone 1 (2.78 gene copy numbers per dry weight), and zone 4 (0.73 gene copy numbers per dry weight) (Fig. 2C). Compared with zone 4, zone 3 increased the abundance of ermB by 2.19 log, zone 2 increased the abundance of ermB by 2.18 log, zone 1 increased the abundance of ermB by 2.05 log. However, no abundance differences occurred between pre and post poultry litter applications (μlog gene copy numbers per gram dry weight in pre-sampling time = 2.86 vs. post-sampling time = 2.45) (Fig. 2B).

There were differences in the abundance of the gene sulI among all three factors, including treatments, sampling time, and zone ($P$ < 0.05) (Fig. 2). For pasture management, the highest abundance was found under CG (μ log gene copy numbers per dry weight = 4.83), followed by RB (4.46), H (4.42), and RBS (2.93) (Fig. 2A). Among zones, the greatest abundance was found in zone 2 (μlog gene copy numbers per gram dry weight = 4.66), followed by zone 1 (4.66), zone 3 (4.38), and zone 4 (2.93) (Fig. 2C). Differences in abundance were identified between pre and post poultry litter sampling time, with a higher abundance of sulI occurring post poultry litter applications (Fig. 2B; log gene

copy numbers per gram dry weight = 4.77 vs. pre-sampling = 4.03). There was also an interaction for pasture management by zone for the *sulI* gene.

Similar to *sulI,* there were differences in the *intlI* gene based on all three factors ($P < 0.05$) (Fig. 2). Based on the influence from pasture management, ANOVA tests indicated that greatest abundances were found under long-term CG (μ log gene copy numbers per gram dry weight = 3.41), followed by H (3.11), and RB (3.07). The least abundance of AMR genes were found in RBS (1.54) (Fig. 2A), with greatest *intlI* gene occurring in zone 2 (μlog gene copy numbers per gram dry weight = 3.22), then zone 1 (3.19), zone 3 (3.18), and zone 4 (1.54) (Fig. 2C). The abundance of the *intlI* gene differed between pre and post sampling, with higher abundances in soil samples collected after poultry litter application (μ log gene copy numbers per dry weight in post-sampling time = 3.79 vs. pre-sampling = 2.28) (Fig. 2B). There was pasture management by zone, pasture management by timing, and zone by timing interactions for the abundance of the *intlI* gene.

After finding differences following long-term pasture management on the abundance of these three AMR associated genes, further analyses were conducted to illustrate the abundance of each AMR associated gene based on the pasture management treatments. Among these 93 samples, *ermB* was found in 77% of samples, while only one amplification was from the RBS (8% of RBS was amplified), and 92 samples were from all treatment groups (85% of treatment samples were amplified). Among these 92 positive samples from treatment groups, it included 78% of RBR samples, 92% of CG, 86% of H, and 8% of RBS. Gene *sulI* was detected in 119 out of 120 samples (99%) and *intlI* were detected in all samples (100%), while $bla_{ctx-m-32}$ was not found in any soils except two (following poultry litter application in zone 2 of CG in 2016 and one in zone 3 of the H treatment in 2017). The gene $bla_{ctx-m-32}$ was not included in the Table 2 and Fig. 2, due to no amplification.

Abundance of these three AMR associated genes indicates there are greater abundances of each AMR associated genes found in CG, RBR and H, relative to RBS ($P < 0.05$; Fig. 2A). For the RBS (no grazing or direct manure or poultry litter deposition), there was no amplification of *ermB* gene from all samples during 2016, while only one sample included an *ermB* gene in 2017. Overall, post poultry litter applications, soil samples had greater abundance of *sulI* and *intlI* genes than pre-application soils. In Fig. 2C, the abundance of these three AMR associated genes were split out based on zones. Based on the factor of zone, differences were observed in these three AMR associated genes, *ermB, sulI,* and *intlI* ($P < 0.05$). Among these four zones, the lowest abundance was found in zone 4 (no cattle manure or poultry litter) among these three genes.

## Distribution of four AMR associated genes in cattle manure and poultry litter

Considering continuous annual applications of cattle and poultry manure were applied to soils (over 14-years), authors were interested in the presence of the four AMR associated genes and whether the abundance varied between soils with manure applied from the two sources. Results from Q-PCR indicated these three AMR associated genes (*ermB, sulI* and *intlI*) were found in all poultry litter samples collected in 2019; however, the abundance of AMR-associated genes from poultry litter in the year of 2018 were below the

**Table 3** **ANOVA of the abundance of three AMR associated genes in cattle manure and poultry litter.** ANOVA results illustrating the differences of the abundance of three AMR associated genes were impacted by the single factor, animals (cattle manure vs. poultry litter), and year (2018 vs. 2019), and interaction between these two factors in cattle manure and poultry litter samples collected from 2018–2019.

| Parameter | Factor | Quantity per gram (log gene copies/gram dry weight manure) $\pm$ SD | $F$-value | P-value |
|---|---|---|---|---|
| *ermB* | Animal (cattle manure vs. poultry litter) | Cattle Manure: 4.66 $\pm$ 0.39 <br> Poultry Litter: 2.45 $\pm$ 0.99 | 6.298 | 0.023[*] |
| | Year (2018 vs. 2019) | 2018: 2.77 $\pm$ 0.64 <br> 2019: 5.08 $\pm$ 0.47 | 8.433 | 0.010[*] |
| | Animal $\times$ Year | | 0.141 | 0.711 |
| *sulI* | Animal (cattle manure vs. poultry litter) | Cattle Manure: 4.60 $\pm$ 0.17 <br> Poultry Litter: 2.40 $\pm$ 1.18 | 6.815 | 0.0189[*] |
| | Year (2018 vs. 2019) | 2018: 3.07 $\pm$ 0.79 <br> 2019: 4.66 $\pm$ 0.34 | 3.452 | 0.082 |
| | Animal $\times$ Year | | 3.893 | 0.062 |
| *intI* | Animal (cattle manure vs. poultry litter) | Cattle Manure: 4.93 $\pm$ 0.52 <br> Poultry Litter: 0.50 $\pm$ 0.50 | 29.524 | 0.001[*] |
| | Year (2018 vs. 2019) | 2018: 2.87 $\pm$ 0.94 <br> 2019: 4.04 $\pm$ 0.85 | 0.865 | 0.366 |
| | Animal $\times$ Year | | 0.461 | 0.505 |

**Note**
[*]$p < 0.05$

detection threshold. The gene of $bla_{ctx-m-32}$ was not found in any poultry litter and cattle manure samples (Table 3). Based on the ANOVA, there were differences in these three AMR associated genes between cattle manure and poultry litter, with greater abundances occurring in cattle manure than poultry litter (53, 95, and 100% greater mean value of gene copies per gram dry weight for *ermB*, *sulI*, and *intlI* in cattle manure than poultry litter, respectively) ($P < 0.05$) (Table 3). The impact of sampling year on the abundance of *ermB* gene was found ($P < 0.05$), with greater abundance in 2019 compared to 2018.

## Prevalence of antimicrobial resistance genes based on pasture management

Purified genomic DNA extracts from six soil samples were chosen for shotgun metagenomic sequencing to evaluate the impact of pasture management on AMR genes. Several unique AMR genes per treatment were identified (Table 4). The number of unique AMR genes, mechanisms, and classes identified in H was lower than other treatments, including the RBS.

Resistome Analyzer in AmrPlusPlus pipeline provided four levels of annotation database hierarchy, at levels of gene, group, mechanism, and class. In each level, the counts of each gene can be found in the output file (Table S1). These identified resistance genes were listed from the greatest numbers of hits (multi-drug resistance class) to least (Bacitracin). Based on the database of MEGARes 2.0, the multi-drug resistance was defined as genes and mechanisms that cause resistance to two or more different antibiotic classes. Typically, such mechanisms involve active extrusion of antibiotic molecules from the bacterial cell

**Table 4  Resistance genes in three features (gene level, mechanism level, and class level).** The number of genes, mechanism and class is the total number of unique AMR gene found without duplication. The gene hits are used as a count of how many times a given gene is found in the data. $n = 6$ soil samples (one replication of CG and H, with two replications for RBR and the RBS at a consistent landscape position landscape position, and sampling timing (i.e., zone 3 and post poultry litter applications)).

| | CG | H | RBR | RBS |
|---|---|---|---|---|
| AMR Gene Number | 210 | 105 | 208 | 143 |
| AMR Gene Mechanism Number | 55 | 42 | 53 | 40 |
| AMR Gene Class Number | 18 | 16 | 19 | 17 |
| Hits | 308 | 139 | 312 | 192 |

or mechanisms that prevent the drug from reaching its target (*Lakin et al., 2017*). The class of multi-drug resistance genes were identified as greater than other resistance gene classes among all treatments and RBS (Table S1). Figure 3 shows relative proportion of hits in each class level by treatments. Overall, the greatest (i.e., 33%) of identified genes from the multi-drug resistance class were found in the CG treatment, followed by RBR watersheds (28%), the RBS (24%), and H watersheds (15%).

## DISCUSSION

### Distribution of four AMR associated genes in soils based on pasture management, landscape position, and sampling time

AMR is a naturally occurring phenomenon, and soils are considered a reservoir for AMR genes (*Kieser et al., 2000*). Overall, pasture management (CG, H, RBR, and RBS) had an effect on three AMR-associated genes, *ermB, sulI*, and *intlI* ($P < 0.05$). Greater abundance of these three AMR associated gene in soils were detected in treatments receiving either long-term poultry litter or cattle manure inputs (CG, H, and RBR), while lower AMR gene abundances were found in RBS, which was not grazed and did not receive direct poultry litter applications. This indicates these three AMR-associated genes were potentially transmitted via animal feces and may be transferred into the soil through animal movement and land application. This finding was consistent with other studies which found that the repeated application of animal manure increased antibiotic resistance genes in agricultural soils (*Luby, Moorman & Soupir, 2016*; *Kim et al., 2017*).

The sample collection timing (pre or post poultry litter application) had an effect on two AMR-associated genes, *sulI* and *intlI* ($P < 0.05$). A greater abundance of these two genes in soils were detected in samples collected after poultry litter applications (July) rather than before poultry litter applications (April). The abundance of *sulI* and *intlI* genes increased after poultry litter applications, indicating poultry litter may include *sulI* and *intlI* genes and increase the abundance of *sulI* and *intlI* genes in soils. This result is consistent with previous work using 16S rRNA sequencing, which found that poultry litter timing greatly influenced soil community structure and gene abundance (*Yang et al., 2019a*; *Yang et al., 2019b*; *Ashworth et al., 2017*). However, poultry litter application timing did not influence the other two genes, *ermB* and $bla_{ctx-m-32}$. Another study pointed out that the concentration of AMR genes (*sulI, intlI*, tetracycline (*tetW*), and streptomycin (*strpB*)) in soil following
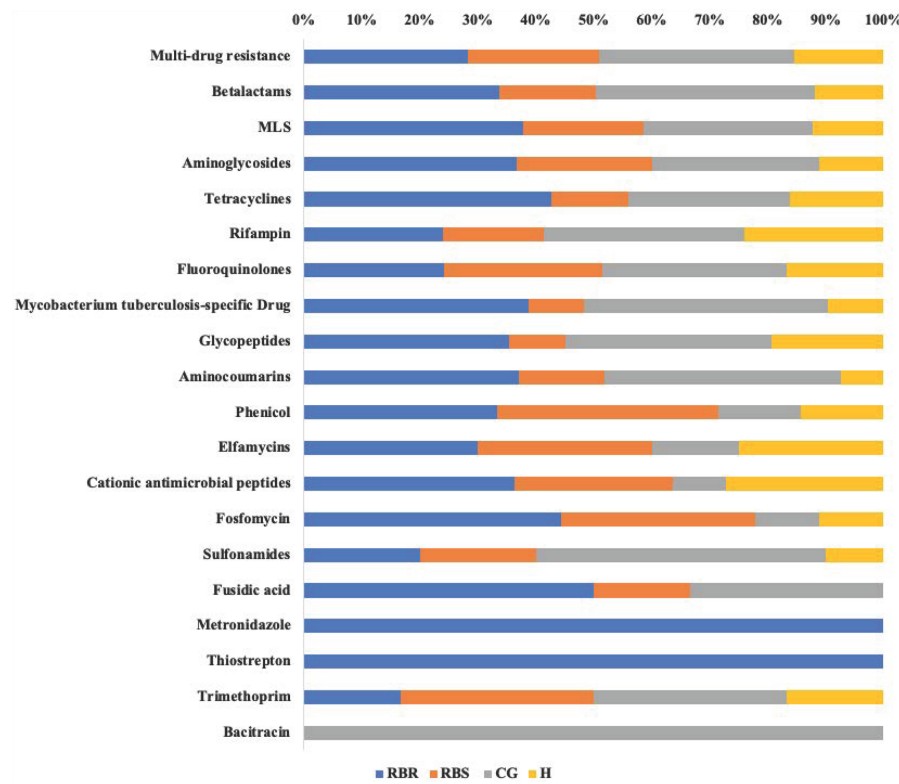

**Figure 3  Relative proportion of AMR genes in grassland soils based on pasture managment.** The relative proportion of AMR resistance gene classes in soils from different pasture management systems. Six soil genomic DNA extracts were sequenced by using shotgun metagenomic sequencing to evaluate the impact of pasture management on antibiotic resistant genes. Each AMR gene classes was normalized to 100% for identifying the percentage of resistance genes from each treatment (CG, H, RBR, and RBS) in each class of resistance gene. Pasture management includes continuously grazed (CG), hayed (H), and rotational grazed with a fenced riparian buffer (RBR). The RBR treatment consists of an additional fenced riparian buffer strip (RBS) that was a non-grazed zone without direct addition of poultry litter or grazing that had trees. Grey=CG, Yellow=H, Blue=RBR, and Orange=RBS.

poultry litter fertilization were greater following 21 to 133 days after application (*Cook, Netthisinghe & Gilfillen, 2014*). These data suggest poultry litter applications may increase the abundance and persistence of AMR-associated genes within the soil.

The factor of zone had an effect on the abundance of these three AMR-associated genes (*ermB*, s*ul* I, and *intI* ) within the soil ($P <$ 0.05), with higher abundance in zone 1, zone 2, and zone 3, and lowest abundance in zone 4. This result suggests that animal manure played an important role in enhancing the abundance of AMR associated genes into the soil. We also found that genes of *sulI* and *intI* existed in the soil of zone 4, even though there was no input from animal manure in this region, thus indicating that these two AMR-associated genes may be inherent to the soil. Since some AMR genes were found in non-agricultural and un-grazed native soils, they were considered as a natural part of pristine habitats (*Durso, Miller & Wienhold, 2012*; *Durso et al., 2016*). This result showcases the importance of evaluating baseline and background levels of AMR genes when investigating the impact
of human input in the occurrence of AMR bacteria and genes (*Durso, Miller & Wienhold, 2012*).

The gene $bla_{ctx-m-32}$ was not detected in most soil samples, indicating that $bla_{ctx-m-32}$ was not prevalent in the locations sampled. Having information on AMR presence in soils is valuable, as previous observations have shown antibiotics may impact the soil microbial community composition and structure, which will ultimately influence ecosystem-scale processes by maintaining these AMR bacteria and genes (*Gutierrez et al., 2010*; *Toth, Feng & Dou, 2011*).

## Distribution of four *AMR* associated genes in cattle manure and poultry litter

Three AMR associated genes (*ermB*, *sulI*, and *intlI*) were more abundant in cattle manure compared to poultry litter. Although, previous studies derived the opposite conclusion. *Wang et al. (2016)* indicated that the *ermB* gene levels in poultry litter were greater than that of cattle manure. Cattle antibiotics and drugs were used over the course of this experiment (Table S2); though, without the information of antibiotics used during poultry production, it is difficult to ascertain that AMR associated genes were related to specific animal management practices. Regardless of where these AMR associated genes originate, we should pay attention to the abundance and movement of these resistance genes, such as *ermB*, because macrolides are a major broad-spectrum antibiotic for human use and play an important role in controlling Gram-positive bacterium infection clinically (*Kanoh & Rubin, 2010*). We also detected three AMR associated genes (*ermB*, *sulI*, and *intlI*) in poultry litter in 2019, but not in 2018. These differing results between years indicates gene presence varied annually perhaps due to differences in environmental or animal management factors. There was a difference identified from the *ermB* gene between 2018 and 2019 with a greater abundance in 2019 and less abundance in 2018. However, it is difficult to ascertain whether this difference was caused by animal inputs without the drug usage information on cattle and poultry in these two years.

## Prevalence of antimicrobial resistance genes from shotgun sequencing following pasture management

Quantifying the prevalence of specific AMR genes may use culture-independent methods, such as Q-PCR, as well as metagenomic sequencing (*Agga et al., 2015*; *Durso, Miller & Wienhold, 2012*). Metagenomic sequencing allows for the tracking of AMR genes and identification of transmission of AMR from animals to the environment (*Oniciuc et al., 2018*). Recent studies using functional metagenomic screening of cattle feces reported the maximum number of AMR genes detected per animal was 26 (*Wichmann et al., 2014*), which was much lower than the number identified from soil samples in this experiment. The identified AMR genes have broad biological activities and might have other functions, rather than only AMR gene encoding. Take the efflux pumps as an example, as it is usually one of the largest AMR mechanisms; however, clinical and laboratory studies suggest efflux pumps have a role in virulence and the adaptive responses as well (*Du et al., 2018*).

The multi-drug resistance gene classes were conferred to phenicol, lincosamide, oxazolidinones (linezolid), pleuromutilin, and streptogramin (*Noyes et al., 2016a*).

However, in the RBS (down slope, but no direct animal input from cattle manure and poultry litter), multi-drug resistance genes were identified. Therefore, it is possible some AMR genes were not from anthropogenic sources, but rather a naturally occurring community component (*Bhullar et al., 2012*), or that surface runoff moved AMR genes downslope. Similarly, *Rothrock et al. (2016)* indicated antibiotic resistant *Listeria* and *Salmonella* spp. occur in all-natural, antibiotic-free, pasture-raised broiler flocks. Future work is needed evaluating the potential movement of AMR genes via surface water runoff (*Jacobs et al., 2019*). *Durso et al. (2016)* also characterized native Nebraska prairie soils that had not been affected by human or food-animal waste products and found that all prairies contained tetracycline and cefotaxime-resistant bacteria, and 48% of soil bacteria were resistant to two or more antibiotics. *Bhullar et al. (2012)* also reported that AMR bacteria and genes can be found from in pristine soil environments that have not been exposed to human antibiotic use, from which, some strains were resistant to a wide range of different commercially available antibiotics. *Cadena et al. (2018)* also reported that tetracycline and sulfonamide antibiotic resistance genes can be identified from organic farming operations.

Based on the MEGARes database, the macrolides, lincosamides, and streptogramins (MLS) A and B were classified as MLS drugs (Lakin et al., 2016), and were identified in soil samples in the present experiment. The MLS class, according to *Tenson, Lovmar & Ehrenberg (2003)*, "contains structurally different but functionally similar drugs acting by binding to the 50S ribosomal subunit and blocking the path where nascent peptides exit the ribosome." *Noyes et al. (2016b)* reported that MLS resistance genes can be detected in both cattle and calves and were equally abundant between dairy and beef herds. Overall, the resistance classes of metronidazole and thiostrepton were identified only from the RBR group, and bacitracin resistance genes were detected only in the CG treatment. Due to the limitation of the sample number for metagenomic sequencing, further studies are necessary to estimate the influence of animal inputs on AMR genes.

## CONCLUSIONS

Results characterized the abundance of AMR genes following 14-years of pasture management using Q-PCR and metagenomics sequencing. The quantitative amplification method suggests increased abundances in three AMR-associated genes (*ermB*, *sulI*, and *intI1*) in soils may be due to long-term cattle manure deposition and poultry litter applications to a lesser extent. Using shotgun metagenomic sequencing, we identified the relative abundance of AMR genes were greater in CG than H, indicating that cattle manure deposition may serve as an AMR source to the environment (relative to poultry litter applications). Additionally, conservation pasture management practices such as rotationally grazing and filter strips decreased soil AMR gene presence, as the unfertilized fenced riparian buffer strip displayed 31.58% lower gene abundance (relative to the CG treatment, based on the AMR gene numbers identified through metagenomic sequencing).While the metagenomic approach has important applications in investigating AMR genes, it is noteworthy that metagenomic methods do have limitations and results may be affected by incomplete resistome databases. Overall, results illustrate that cattle manure inputs may

influence AMR abundance in soils and conservation management may minimize AMR gene presence in the environment.

### Funding

The authors received no funding for this work.

### Competing Interests

The authors declare there are no competing interests.

### Author Contributions

- Yichao Yang performed the experiments, analyzed the data, authored or reviewed drafts of the paper.
- Amanda J. Ashworth conceived and designed the study, authored and reviewed drafts of the paper, and approved the final draft.
- Jennifer M. DeBruyn and Lisa M. Durso analyzed the data, prepared figures and/or tables, authored or reviewed drafts of the paper, and approved the final draft.
- Mary Savin and Kim Cook reviewed drafts of the paper, and approved the final draft.
- Philip A. Moore Jr. and Phillip R. Owens oversaw the larger field experiment, authored or reviewed drafts of the paper, and approved the final draft.

### Data Availability

16S data are available at NCBI: PRJNA629832.

Soil qPCR data are available at Figshare:

Yang, Yichao (2020): Soil_qPCR Data for Peerj.xlsx. figshare. Dataset. https://doi.org/10.6084/m9.figshare.12315290.v1.

### Supplemental Information

Supplemental information for this article can be found online at http://dx.doi.org/10.7717/peerj.10258#supplemental-information.

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
