# Peer review of "Antimicrobial resistant gene prevalence in soils due to animal manure deposition and long-term pasture management"

_PeerJ, doi:10.7717/peerj.10258_

## Round 0.1 · original submission · Major Revisions

We received two very detailed reviews of your submission about AMR genes in grassland systems. Both reviewers found the manuscript scientifically relevant but that major revisions are needed. The text needs revision for style and length. I provide some specific examples below, but this is not a comprehensive list.

Data presentation: 1) Reformat Figures 2 and 3 to jitter, violin or box plots that allow readers to visualize the distribution of data. 2) Present the number or Miseq reads (raw and after QC) 3) Provide SRA ascension numbers in Methods.

Specific examples:
Line 40. Did you look at bacteria or just genes? Looking at bacteria would require culturing or live-dead metagenomic techniques (https://microbiomejournal.biomedcentral.com/articles/10.1186/s40168-017-0285-3).
Line 49. Only Muhammad Ali is “the greatest.” Revise.
Line 53. This sentence needs revision. Here and throughout, use the word “however” carefully and write more succinctly and directly. For example, replace “Additionally, soil sulI and intlI gene abundance increased following poultry litter applications, however, ermB, sulI, and intlI had greater gene copy abundances per gram dry weight in cattle manure than poultry litter” with “Poultry litter had lower abundance of AMR genes ermB, sulI, and intlI relative to cattle manure. Long term application of this litter increased the abundance of sulI and intI genes in soil but the abundance of ermB …” Also, what happened to blactx-m-32?
Line 58. Replace “illustrate” with “suggest”
Line 62. As I understand most antibiotic use in animal husbandry is to increase weight gain not to treat infections. These antibiotic usages are not considered essential (https://www.ncbi.nlm.nih.gov/pmc/articles/PMC4638249/).
Line 63. Here and throughout, write succinctly. Replace “may provide selection pressure for the evolutionary phenomenon known as antibiotic …” with “selects for antibiotic…”
Line 66. Delete “may,” unless you doubt the papers you present as examples.
Line 88. Delete “had”
Line 89 Delete “owing to greater organic animal inputs” (Yang et al., 2019b) and replace “suggesting” with “which suggests”
Line 90 Replace “increases microbiome diversity and may be a mechanism for improved soil health.” with “improves soil health.”
Line 143. Delete “(C2H6O)”
Line 147. Delete “which was used to present data per gram dry weight”
Line 210. Say what the difference is, instead of “There were differences in soil ermB gene detection among the treatments.” See also lines 221 and 231.
Line 284. Here and throughout, delete “in order”
Line 386. Delete “work from”
Line 392. How are “soil samples resistant”?
Lin 416. Conclusions should be one paragraph and not repeat previous sections (Methods, Results..)
References do not follow a consistent pattern. I provide examples below but this is not a comprehensive list. Please identify an author that is specifically responsible for proofreading references.
Line 635. Provide vol:page-page.
Line 640. Either period after abbreviations (Proc.) or not (Proc) consistently.
Line 647l Why is this journal not abbreviated but most others are.
Line 740 Use a consistent format for all references (Soil Biol. Biochem. 43, 2470 –…) should be 43:2470..). See also 751 and 756.
Line 747. Why cap “Antibiotic”
Line 755. Only cap proper nouns in article titles.
Line 756. Vol: page-page only (no issue)
Line 758 No period after Mbio see also 764.

Reviewer 1 ·

Basic reporting

The aim of this study was to determine how the combination manure and litter application along with 3 different grazing procedures affects AMR gene levels in soil. The study is well justified in that there are few studies highlighting the effects of pasture management on AMR gene abundance. The figures are mostly effective, but a few modifications are necessary. The sequences are available on NCBI based on the links given on the website, but the links and accession numbers are not in the text.

Fig 1 explains the experimental design clearly and was helpful to reference while reading through results.
Fig 2 is also well labeled, except number of samples for each mean abundance (n) should be specified. It would be preferable to achieve a similar resolution as Fig 1 and 3, where there’s no obvious pixelation.
Fig 3 is clear, but it could be a helpful reminder in the legend to note how many of each of the soil samples (n) were used for shotgun sequencing.

Experimental design

The use of 3 different pasture management practices allows for a comprehensive study and were described well, especially in fig 1. Sampling the soils and the manure and litter was informative, but would have been stronger if performed in the same year. A few more details in the stats and shotgun metagenomics are needed for clarification.

Lines 183-185: Was Tukey’s HSD also applied for multiple comparisons?
Lines 188-191: For clarification, which time point was chosen for CG and H, and was a pre- and post- application soil sample chosen for RBR and RBS each? It could be clearer to just list explicitly what the 6 samples are (for example, post-application zone 1 CG, post-application zone 1 H, post-application zone 1 RBR, etc.)
Lines 200-202: Is a positive hit 85% of the query read aligns with a subject AMR gene, or is a positive hit 85% of the subject AMR gene aligns with the query reads? Does the 85% refer to coverage or identity? What is the minimum length of a read?

Validity of the findings

There is an interesting story in the results, especially with the interaction effects, but more statistical methods need to be described to verify the analyses for the QPCR data. For example, how were the multiple comparisons made, and what were the detection limits for each of the genes? For values not detected for certain genes in the samples, were those values excluded, or were they substituted with a number at or below the detection limit?

Lines 215-216: It would be clearer to note the units after the numbers. Rather than listing the mean log gene copy numbers measured from all the treatments, it could be more informative in the text to state by how many logs the grazing treatments increase the ermB compared to RBS. Same comment with lines 217-218.
Line 219-220: If there are no significant differences between treatments, it is necessary to state the means or log difference of those treatments.
Lines 233-235: These statements as they are structured appear like CG has significantly the greatest abundance of the 3 genes compared to the other treatments, but then the figure has only the star to indicate that RBS is significantly the least abundance of the genes. If CG is also significantly greater in gene abundances than the other grazing methods as well, it would be clearer to change the stars in the figure to connecting letters. Or if it’s a more interesting story, the text may also be reworded to note that the grazing methods that were coupled with manure and litter application were all significantly greater in gene abundances than RBS by over x log. This comment applies to all 3 genes and the other 2 factors.
Lines 242-244: This paragraph describes detection and non-detection more than the abundance of each gene. This paragraph would be more useful in the beginning of the results to help give context of how many samples were used for comparisons by ANOVA.
Lines 245-247: Remove this sentence fragment and just skip to “Among these 93 samples…” and then use the absolute number of samples instead of the percentages in the list.
Line 250: change “ collected following” to “, one following”.
Line 251: insert “one” between “and” and “in”.
Lines 253-265: This paragraph is very interesting in that it highlights the main story told in the Q-PCR data, but it also seems redundant. If the first 3 paragraphs could be reframed as suggested in an earlier comment, this paragraph could be excluded.
Lines 280-281: How were the abundances of sul1 and intI1 not significantly different between years when they were detected in one year but not the other? The statistics section of the methods also needs to have a statement of how non-detectable values were incorporated into the analysis if this is the case.
Lines 285-289: These numbers are already listed in the table, so these sentences can be omitted.
Lines 323-324: In general, specific CTX-M genes can be difficult to detect using PCR methods since many similar genes are hybrids of 2 or more other CTX-M genes. In the future, Birkett et al. 2007 could be a good resource for detecting a range of CTX-M genes.
Lines 342-345: This citation is a useful reference when discussing the importance of many classes of AMR genes in soil and the connection to human pathogens, but it does not connect well to the point that blaCTX-M-32 was not detected in soils in this study. Are there other studies that point out blaCTX-M-32 in soils or manure?
Line 392: Bacteria from soil samples?
Line 423: While interesting to point out, this is not pointed out in fig 2, but using letters instead of the stars would help depict this. Did the RBR practice also result in greater gene abundance than H as well since cattle manure was used here too?
Table S1: The legend appears missing.

Additional comments

The use of “AMR genes” and “ARGs” seems to be used interchangeably, but it looks like “AMR genes” is used more often, so for consistency, use only one term.

Reviewer 2 ·

Basic reporting

- Line 30: typo on Antibiotic resistance- please fix.
- Line 40 (Abstract): What type of residues? If not focusing on this in the study, it would probably be better to remove, or if not, to clarify what type of residues the authors are referring to.
- Line 58-59: Saying as a conclusion in the abstract that poultry litter may minimize AMR genes might be a stretch given what this study showed. I would suggest removing that idea.
- Line 64: AMR is defined as antibiotic resistance, but usually AMR is referred as antimicrobial resistance. I would suggest defining AMR as antimicrobial resistance.
- Line 71: on what type of soil was that application? I would suggest adding that information.
- Line 72: Please add a citation for 'ARGs can be found naturally'.
- Line 79: un-degraded is one word: undegraded.
- Line 83: Here, the authors talk about poultry litter for the 1st time but there were no previous studies in the background/Intro section that referred to poultry litter and ARGs. It would be nice to have some context about poultry litter the same way the other manures were mentioned.
- Line 97: any specific manure? If so, I would suggest being specific in the aims about the type of manure the authors evaluated.
- Line 643 (References): typo on the reference re: the word 'antibiotics'.

Experimental design

- Lines 105 and then line 109: It is a bit confusing to first read the experiment had 9 watersheds and then refer to only 3 watersheds in line 109. I would suggest clarifying this.
- Line 115: Please provide a reference to 'the best management strategy'.
- Lines 119-123: This portion is a bit confusing. I would suggest either removing some of it if it is not relevant, or adding a bit more detail to the figure to show what it means by shoulder, upper, etc.
- Line 127: Did this commercial farm give antibiotics to the broilers? That would be good to mention, and if known, it would be ideal to mention which specific antibiotics.
- Line 131: I would suggest removing the plural in 'collections'. The sample collection section could be reworked a little bit to be clearer about the type of samples, timing, and number of samples of each type.
- Lines 135-136: Please clarify if the soil samples were taken manually or with a robot of some sort.
- Line 140: Please clarify what the authors mean by in-house piles.
- I am a bit confused about the timing of sampling of poultry and cattle manure. Why was the cattle manure collected in 2018-2019 and poultry in 2016-2017?. Also, it is unclear how many times cattle manure samples were collected.
- Line 150: when the authors say "cattle manure, or poultry litter..."- so not all samples were analyzed the same way? Or should that 'or' be 'and'?
- Why were these four genes in particular chosen? Some explanation related to the decision behind choosing these genes would be good.
- Line 165 is redundant. I would suggest removing it.
- I appreciated the authors giving lots of details on the steps from the qpcr and the manipulation of the data. It is not as common to find all these details. This makes it more transparent and reproducible.
- Line 184: is the log transformed data referring to each of the genes? I would suggest explaining this/expanding on it. Was each one of the 4 genes analyzed used as a dependent variable in the analyses, or were the 4 genes analyzed altogether?
- Line 189: Why were those 6 samples chosen for the metagenomics analyses? Some explanation about this would be good.

Validity of the findings

- The results section is a bit long. I would suggest the authors trying to summarize the results further. Some suggestions below.
- Figure 2 has low resolution- the text is a bit blurry. Also, I could not see in the figure the A, B, C for each panel of figure 2. Also, I am a bit confused about the quantities mentioned for each gene and each part of the figure. In the tex, the quantities are referred to as micro log gene copy numbers per dry weight while in the figure the y axis is log only and it is hard to know the quantities exactly only inferred by the figure. Please clarify and be consistent with the text and figure.
- Lines 257-258: "The gene...": Please reword/fix this sentence.
- The paragraph that goes from line 253-265 can probably be integrated in the previous results. Some of the information is redundant and it makes the results too long.
- Supplementary Table 1 would need a repeated heading as it extends beyond one page.
- Line 271: I would suggest removing the word 'these'.
- Line 277: Here please clarify the units again as mentioned before for the micro gene- is this log or gene copies and in micro?
- Line 296: How was multi-drug resistance defined? How many resistance genes?
- Figure 3. Please capitalize betalactams to be consistent with the other headings.
Discussion:
- Line 305: Please rephrase "AMR genes is an evolutionary phenomenon" to something like " AMR, including AMR genes is a naturally occurring phenomenon", something along those lines.
- Line 325: add ' of' before the word ' AMR genes.
- Line 326: add either 'to' or 'in' before 'fertilized soils'.
- Line 343-344: I do not understand this sentence. There seem to be two ideas- finding the ctx-m and beta-lactamase genes and that they might be the same as the ones found in human pathogens. But finding these in soil does not necessarily imply they are in human pathogens. Please clarify the ideas.
- Somewhere in the intro and/or discussion it would be good to mention if these experimental watersheds receive the input potentially from other sources- human or other animals, etc.
- Line 354: Please also make a point to comment about the use of antimicrobials in your experimental design under your comparison sites and when talking about the animals.
- Line 361: remove the word 'that'.
- Line 361- 362: Given only 2 years of data, I would suggest not making strong conclusions about the differences related to the temporal component. I think more years would be needed to compare that factors.
- Lines 376-380: Please rephrase this section.
- Lines 390-397: This paragraph should be better integrated into the results of the experiment. It seems like floating there with two different ideas but are not well connected to the previous paragraph.
- Lines 400-403. Please connect the idea about the use of beta-lactam antibiotics and the finding of beta-lactamase gene in your experiment., As it is, it reads as two different ideas not well connected.
- Line 416: change 'characterize' for 'characterized'.
- Line 418: remove the word 'these'.
- Line 421: This idea of live and dead bacteria is the 1st time it appears. I would suggest talking about this during the discussion, or removing it if not relevant.
- Supplementary Table 2: Please try to make the table more consistent. For example, Reason for the drug used includes deworm, parasites- what does it mean parasites vs deworm?; for Amount, also include route of administration and duration if that information is available.

Additional comments

This study presents an interesting contribution related to the potential dissemination of antimicrobial resistance (AMR) genes in the environment, specifically in soil. There is still a need for more studies evaluating AMR in soil. I have made specific suggestions and comments for this section. Overall I feel the paper can be more concise. I think once the concerns have been addressed, this paper will be worthy of publication and will add an important contribution to the field.

---

## Round 0.2 · Minor Revisions

I agree with the reviewers that Figure 2, while an improvement, needs work. I suggest defining the boxes in the figure, so the reader can identify the treatments. Also, use a consistent format for journals in references. Some are not abbreviated (line 450), some are with a period (line 463) and some without (line 466).
Line 474. This reference is incomplete.
Line 517. Only capitalize proper nouns in titles (not Antibiotic). See also line 588.

Reviewer 1 ·

Basic reporting

The grammar improvements have made this manuscript a smoother experience to read, and their conclusions indeed make sense given their data and past research. Most of the suggestions here are minor, but figure 2 still needs improvement.
Line 100: insert “to” before “balance”.
Line 104: insert “to” before “evaluate”.
Figure 2: The boxplots are much more informative than the bars from before. However, differentiating between the stars indicating significance and the dot plots that indicate the outliers is difficult visually. It would still be more informative to include the exact results of the Tukey’s differences in the figure. JMP is usually pretty clear with providing connecting letters to indicate exactly which pairwise comparisons are significant, and the letters would also be easier to see than the stars.
Table 3: For the 3rd column, is this the mean quantity ± standard deviation?

Experimental design

Lines 149-150: When was poultry litter applied? Just wondering how much time passed between application of litter and sampling.
Lines 183-184: Thank you for including how you worked with values below detection. What were the detection limits per gram of soil?
Lines 208-210: Thank you for indicating that the 85% referred to coverage. For the queries that aligned with the subject AMR genes, did they have a minimum %identity?

Validity of the findings

Line 211: “pre-“ and “post-“ should replace “pre” and “post”
Line 212: I would change “detection” to “abundance” since ANOVA is really comparing the means between the pre- and post-litter application, so all the values that the test is comparing are detected
Line 248: I would change “abundance” here to “detection” in this case, or “proportions of detected to non-detected samples”
Line 302: insert “,” after phenomenon
Line 336: Change “found” to “detected”
Line 343-344: I would just say ctx-m-32 wasn’t detected rather than there was no difference between the manure and litter, or even omit that fragment since it was already stated in the paragraph above
Lines 361-362: It would be preferable to reword “relies” here since it’s not the only way to characterize AMR genes. Culture-independent methods are vital for many AMR genes from organisms that are hard to culture, but culture-dependent methods also have importance in AMR research (McLain et al. 2016, JEQ)
Lines 377, 379-380: Jacobs et al (2019, JEQ) could be a valuable reference here regarding AMR gene movement via storm water runoff

Reviewer 2 ·

Basic reporting

I feel the authors improved their background and justification of the paper. It now reads more clearly.

Minor comments:
- Line 64: typo before the word 'fundamental'- a instead of an.
- Line 101: I think there is a missing word between effort and balance. Please check.
- Line 115: Materials and Methods
- Line 204: it would be"using Tukey’s honestly significant difference test"
- Line 220: Correct Bos taurus
- Line 271: Please correct/modify the x before zone. Same for lines 283-284. Please express this using words instead of the x.
- Line 314: typo in applications
- Line 318: add 'the' before lowest

Figure 2. I would suggest saving this figure in higher resolution to see better (unless this was something from the pdf review and it has better quality than what it looks like). Also, please include either a legend about the colors used that differentiate each group in each panel (A, B, C) or else add what the colors mean to the figure legend text.

Figure 3. Please define acronyms like MLS in the legend.

- Line 438-439: Please try rephrasing this sentence to remove "Take the.."
- Line 453: Please check the 'from' and 'in' together.

If possible, it would be nice to see in the suppl. materials the specific AMR genes that were found through sequencing for each antibiotic class, or a link to the data.

Experimental design

The research question and goals are more clearly defined. The methods read much better now, and I have no further concerns for this section.

Validity of the findings

No further comments.

Additional comments

The authors improved the manuscript based on the reviewer's comment, and I understand how much work this is, so I have only minor comments for the authors at this point.

---

## Round 0.3 · Minor Revisions

This manuscript will be accepted as soon as these few minor revisions are addressed.

Reviewer 1 ·

Basic reporting

Figure 2 looks much clearer, and I believe readers will greatly appreciate the more details added throughout the manuscript.

Table 2: If possible, the format at the end of the table might need to be fixed. Line 17 is separated.
Figure 2: Lines 432-436, double check the legend reflects the changes. “star” should be changed to “connecting letters”
Overall, double check the formatting and ordering of the figures and tables and their legends at the end of the pdf

Experimental design

Line 204: It would be helpful to provide the URL to the Resistome Analyzer tool

Validity of the findings

No further comments

Reviewer 2 ·

Basic reporting

No comment

Experimental design

No comment

Validity of the findings

No comment

Additional comments

I think the authors made nice improvements to the paper and this will be a nice contribution to the field.

---

## Round 0.4 · Minor Revisions

The manuscript appears scientifically sound and is accepted but I have a few suggestions for style and format that should be addressed before we can enter production.

Line 39. I find the phrase “especially in the US’ largest agricultural land-use or grasslands” awkward and the presence of AMR genes in manure is established. I suggest revising to “especially in grasslands. Animal manures are spread widely on grasslands, which are the largest agricultural land-use in the United States. These nutrient-rich manures contain AMR genes. The aim...”

Line 50. Delete “the” in “the highest” and using the word “following” twice is awkward. Revise to “were highest (P < 0.05) in samples collected from continuously grazed plots, which suggests overgrazing increases AMR gene persistence.”

Line 75. Replace “However, it should also be noted that AMR genes can be found naturally..” with “However, AMR genes occur naturally..”

Line 499 & 505. Be consistent in journal titles. Either spell them out (Current Opinion in Microbiology) or not (Crit. Rev. Microbiol) and if the title is abbreviated use a consistent format. For example, some references use a period to indicate an abbreviation (Environ. Sci. and Tech.) and some do not (Clin Microbiol Rev). Further, some use the symbol “&” (Environ. Sci. & Techn.) and some do not.

---

## Round 0.5 · accepted · Accept

I appreciate your careful consideration of the reviewer's comments.